# Implications for Immunotherapy of Breast Cancer by Understanding the Microenvironment of a Solid Tumor

**DOI:** 10.3390/cancers14133178

**Published:** 2022-06-29

**Authors:** Alexander S. Franzén, Martin J. Raftery, Gabriele Pecher

**Affiliations:** Competence Center of Immuno-Oncology and Translational Cell Therapy, Department of Hematology, Oncology and Tumorimmunology, CCM, Charité-Universitätsmedizin Berlin, Berlin Institute of Health @ Charité, 10117 Berlin, Germany; alexander.franzen@charite.de (A.S.F.); martin.raftery@charite.de (M.J.R.)

**Keywords:** immunotherapy, breast cancer, tumor microenvironment, cancer-associated fibroblasts, tumor-infiltrating lymphocytes, tumor-associated macrophages

## Abstract

**Simple Summary:**

Breast cancer is one of the most common forms of cancer in women. Treatment options include immunotherapy where elements of the immune system are used directly or in a modified form. However, the surrounding ecosystem of a solid tumor forms a complex barrier of supportive and protective cells that needs to be penetrated to allow immunotherapies to be effective. A better understanding of the underlying mechanisms of the tumor microenvironment will help improve immunotherapies. This review will summarize the latest research concerning the tumor microenvironment of breast cancer and give implications for immunotherapy.

**Abstract:**

Breast cancer is poorly immunogenic due to immunosuppressive mechanisms produced in part by the tumor microenvironment (TME). The TME is a peritumoral area containing significant quantities of (1) cancer-associated fibroblasts (CAF), (2) tumor-infiltrating lymphocytes (TIL) and (3) tumor-associated macrophages (TAM). This combination protects the tumor from effective immune responses. How these protective cell types are generated and how the changes in the developing tumor relate to these subsets is only partially understood. Immunotherapies targeting solid tumors have proven ineffective largely due to this protective TME barrier. Therefore, a better understanding of the interplay between the tumor, the tumor microenvironment and immune cells would both advance immunotherapeutic research and lead to more effective immunotherapies. This review will summarize the current understanding of the microenvironment of breast cancer giving implications for future immunotherapeutic strategies.

## 1. Introduction

Breast cancer is one of the most common cancers worldwide, with a global yearly estimate of 2.3 million new cases and 685,000 yearly deaths, making breast cancer the fifth leading cause of cancer mortality worldwide [1]. Breast cancers are commonly categorized by their histopathological appearance (carcinoma in situ and invasive carcinoma) and molecularly by the expression of estrogen receptors (ER), progesterone receptors (PR) and human epidermal growth factor 2 (HER2) receptor. The molecular classification is further subdivided into luminal A (ER+, PR+ and HER2-), Luminal B (ER+, PR+ and HER2+), HER2-enriched (ER-, PR- and HER2) and basal-like (triple negative breast cancer ER-, PR- and HER2-) [2]. Receptor-positive breast cancers can be targeted by their molecular markers since they are dependent on these for growth. Common therapies targeting breast cancer include neoadjuvant therapies such as hormone treatments, chemotherapy and immunotherapy before more radical treatment regiments such as mastectomies are used [3]. Immunotherapy is a form of therapy that takes advantage of components of the immune system such as antibodies and administers these as a treatment against cancer [4]. In breast cancer, current immunotherapies are focused on antibodies targeting molecular receptors on the breast cancer cell surface or targeting the tumor-infiltrating immune cell subset in the tumor microenvironment. The former treatment leads to receptor blockade inhibiting proliferative pathways, such as HER2 [5]. The latter treatment makes the immune cells more potent in finding and eliminating tumor cells or preventing the immune cells from being inactivated by the tumor cells [3,4,6]. Moreover, a growing body of research is focusing on the development of cellular immunotherapies involving genetically modified anti-tumor targeting immune cells, giving them the ability to destroy and hone in on molecular structures on cancer cells [7]. Cellular immunotherapy has shown spectacular success against some forms of lymphoma. However, the success against solid tumors have been more limited [7,8]. The tumor microenvironment (TME) of a solid tumor typically generates an area of immunosuppression and inflammation around itself that promotes growth and protects the growing mass. Instead of attacking the neoplasm, elements of the immune system are coopted by the tumor [9]. In order to attack the tumor, immunotherapies must penetrate this penumbral region without becoming inhibited, and at the same time not show off-target activity distal to the tumor. The response to cancer therapy can be traced back to the patient’s own tumor microenvironment (TME) and its components [10,11,12]. The TME is a complex part of a solid tumors’ formation and includes a multitude of components and processes that aids and supports the tumor growth [9,12]. Due to its complexity, the TME is often referred to as its own organ consisting of cancer-associated fibroblasts (CAF), infiltrating immune cells such as tumor-infiltrating lymphocytes (TIL) and tumor-associated macrophages (TAM) [9,13,14,15]. We have summarized the current understanding of the breast cancer tumor microenvironment and its main constituents.

## 2. Breast Cancer Microenvironment

In this review we will focus on the best understood components of the breast cancer microenvironment, the CAF and the infiltrating immune cell subsets, TAM and TIL (Figure 1).

### 2.1. Cancer-Associated Fibroblasts (CAF)

Fibroblasts are the principal component of connective tissue and are responsible for the synthesis and modulation of extracellular matrix [13]. In the tumor stroma, a special type of fibroblast generally referred to as cancer-associated fibroblasts (CAF) are found. It is thought that CAF are key sources of proteolytic enzymes, growth factors and cytokines that drive tumor progression through ECM (extra cellular matrix) remodeling, inflammatory and proliferative signaling [9,13,16]. A clear definition of a general CAF population does not exist due to both the heterogeneity of the different tissues where CAF are present as well as the multiple cell types that can form CAF, including locally residing fibroblasts, pericytes, adipocytes, endothelial cells or bone marrow derived mesenchymal cells [16]. The current consensus among CAF researchers is that the majority of CAF probably originate from activated local tissue-resident fibroblasts, but there are clear examples demonstrating alternative origins [16].

CAF are distinguished by an elongated morphology and typical cell surface markers that occur when these cell types encounter tumor tissue [9]. However, because of the large heterogeneity seen in CAF, they are mostly defined by a combination of markers, functionality, spatial location and morphology [16]. Principle CAF markers include alpha-smooth muscle actin (α-SMA), fibroblast specific protein 1 and fibroblast activating protein (FAP) [17]. Additional markers include, alpha/beta (PDGFR α/β) and passive markers, such as an abundance of ECM components (collagen 1 and fibronectin) [9,13,17].

Growth factors promoting CAF formation include tumor growth factor beta (TGF-beta), Fibroblast growth factor-2 (FGF2) and PDGF [16,17]. Of special importance from an immunological perspective is TGF-beta, as it inhibits CTL and NK cell activity [18,19]. However, the mentioned markers and growth factors are not exclusive to CAF and when interpreting results linking to CAF, a holistic view of the tumor stroma is paramount.

Clinical evidence has linked fibroblast-induced high mammographic density to an increased lifetime risk of developing malignancies, suggesting fibroblast involvement in driving disease progression in breast cancers [20,21]; however, more evidence needs to be gathered to confirm this hypothesis.

When characterizing the stroma of breast cancer patients, Priwantigrum et al. found an abundance of the CAF marker α-SMA and high densities of collagen 1 located next to tumor nests in the breast cancer tissue, indicating CAF involvement in tumor progression. Interestingly, the results also showed an abundance of blood vessels in the stromal regions of the tissue samples which implies that drugs administered need to pass a stromal barrier when extravasating to reach the tumor cells, suggesting a tumor protective physical barrier [22]. This is in line with findings from Kaukonen et al., where a clear difference was found in the cell-derived matrix stiffness between CAF and normal fibroblasts [23]. A stiffer ECM was produced by CAF and an increase in the proliferative rate of triple negative breast cancer cells when grown on a stiffer ECM was observed [23]. Furthermore, it has been shown that the stroma in aggressive breast cancer subtypes is stiffer with more linearized collagen bundles, especially at the invasive front of aggressive subtypes (basal like and Her2+) compared to less aggressive subtypes (luminal A and B) [21,24]. This finding also showed a larger concentration of TGF-beta and infiltrating M2-type macrophages in the invasive front of these aggressive breast cancer subtypes [24]. A stiff ECM produced by CAF could therefore not only serve as a protective physical barrier, but it could also play a part in driving tumor progression and tumor malignancy [22,23].

Early experiments have revealed that culturing fibroblasts in tumor conditioned media or with TGF-beta leads to an activated form of fibroblasts. Yu et al. showed that isolated CAF from primary breast cancer tissue induce a more aggressive behavior in breast cancer cell lines when co-cultured or treated with conditioned CAF media [25]. Similar findings were made in a mammary 3D model with the focus on cell–cell interaction between dermal fibroblasts, mammary fibroblasts and the breast cancer cell line MDA-MB-231. A faster aggregation between fibroblasts and tumor cells was observed when fibroblasts were pre-conditioned with tumor cell media prior to co-culture. Furthermore, a faster aggregation was observed between human primary mammary fibroblasts and breast cancer cells compared to dermal fibroblasts, indicating a form of tissue specificity [26]. It has also been shown that the tumor secretosome and the capacity to convert mesenchymal stromal cells to CAF differs between breast cell lines, and that this was linked to different phases of tumor progression [27]. This is in line with the accumulating body of evidence pointing towards CAF being a heterogeneous population of cells with overlapping markers depending on tumor type, stage and origin. Further bolstering this claim is research reported by Raz et al., showing that bone marrow mesenchymal stem cells (BM-MSC) are a substantial source of CAF in breast carcinomas and that they are specifically recruited to breast cancer tumors. By injecting a mouse model with traceable BM-MSC, they could demonstrate that BM-MSCs in circulation only started to differentiate into CAF once they reached the tumor tissue [28]. This indicates that bone marrow-derived CAF are recruited to primary tumors and metastatic lesions and express different markers than locally residing CAF, implicating CAF heterogeneity [28].

A detailed analysis of CAF in breast cancer by Costa et al. showed four distinct CAF subsets (S1–S4) in the breast cancer stroma. Each subtype showed distinct properties and accumulated either within tumors (S1 and S4) or juxtatumorally (S2 and S3). Interestingly, CAF-S1 content was linked to an increased infiltration of T-reg cells and macrophages while inversely correlating with CD8+ infiltration. Furthermore, it was demonstrated that CAF-S1 enhance T-reg differentiation, activity and inhibition of effector T cell proliferation. These findings show an immunosuppressive role of the CAF-S1 subtype which could not be confirmed in the other fibroblast subtypes [29].

Development of CAF-specific therapy has been complicated by the expression of many of the CAF markers on normal tissues. Suggested strategies being developed to suppress CAF in the TME include controlling key CAF signaling molecules such as TGF-beta in the tumor microenvironment, CAF depletion through cellular immunotherapies or oncolytic viruses, ECM targeted therapies and reprograming CAF into tissue resident fibroblasts [16,30]. Some studies have been able to specifically target CAF in the TME. By creating an oncolytic adenovirus expressing a bispecific T cell engager targeting both fibroblast activation protein and CD3e, it was possible to reverse CAF inhibition [31]. In an alternative approach using a dual chimeric antigen receptor (CAR)-T treatment for multiple myeloma targeting CAF and myeloma cells, it was possible to overcome CAF inhibition of CAR activation [32].

Taken together, CAF are a multifaceted component of the TME and as the most abundant cell type in the tumor stroma, they provide the tumor with proliferative support and protection. To overcome the tumor stroma in solid tumor immunotherapy, a multitude of factors needs to be considered and targeted, such as immunosuppressive signaling, physical stroma barriers, the reciprocal signaling between CAF, tumor and supporter cells and the re-modulation of the surrounding ECM driven by CAF.

### 2.2. Tumor-Infiltrating Lymphocytes (TIL)

TIL are generally defined as tumor-infiltrating lymphocytes [33]. The density and diversity of TIL in the breast cancer TME are closely related to prognosis and response to immunotherapy [34,35]. In breast cancer, the most abundant lymphocyte subtypes that are found are CD4+ and CD8+ T cells [36,37]. Thus, the literature and immunotherapy strategies concerning TIL in breast cancer are mostly focused on T cell subtypes. Tumors that have a high CD4+ and CD8+ T cell count are considered immunologically “hot” and it has been demonstrated that these tumors respond well to immunotherapy. In contrast, immunologically cold tumors with low T cell infiltrate respond poorly to different forms of immunotherapy [15]. There is a clear association between the number of TIL and the response to neoadjuvant therapy or to checkpoint inhibition [34,35]. In some subsets of triple negative breast cancers, the TIL are an adverse prognostic factor, implying that the makeup of the TIL is important for the development of the cancer and is linked to the cancer type [38].

Breast cancer tumors with high amounts of T-regs (CD4+ FOXP3) have a poor prognosis as the T-regs exhaust the local T cell immune response [39,40]. However, this exhaustion might be counteracted by immunotherapeutic strategies that aim to redirect the T cell response with antibodies that bind cancer antigens to facilitate effector cell and cancer cell interaction. Research from Egelston et al. showed that exhausted T cell populations from breast cancer tumors could be redirected and reactivated by using bi-specific antibodies, which specifically bind to two different antigens. In contrast, exhausted T cell populations from melanoma patients did not show the same ability to reactivate after exhaustion [41]. This has implications for immunotherapeutic breast cancer treatment regiments in which local residing T cells could either be redirected to attack tumor cells using bi-specific antibodies or could be utilized in a combination approach using adoptive cellular immunotherapies. Combination therapy approaches in which an adoptive cellular immunotherapy utilizing engineered immune effector cells in combination with antibodies show a lot of promise in blood malignancy research, but there is a need for more clinical trials in a solid tumor setting [42,43].

The role of B-cells and plasma cells in anti-tumor immunity is disputed but their involvement in the TME of breast cancer have recently been established [44,45,46,47]. Recently, a high intertumoral plasma cell density was associated with longer disease-free survival and longer time to relapse in TNBC suggesting a humoral response in TNBC [45,47]. However, the underlying causes of these findings is still in need of further investigation.

The development of single cell techniques has made it possible to analyze the complex cancer ecosystem that exists between the immune cell landscape, the tumor cells and the tumor microenvironment. Using single cell RNA-sequencing, Azizi et al. established that there is a greater phenotypical diversity between immune cells found in the tumor tissue compared to immune cells found in normal mammary tissue. By measuring the variance of activated genes such as IFNα, IFNγ, TNFα and TGFβ present in the tumor tissue and in normal mammary tissue, they defined a metric denoted “phenotypic volume”, which was used to characterize the phenotypic expansion of immune cells in the tumor microenvironment. A significant increase in the “phenotypic volume” across all major immune cell types, such as T-cells, NK cells and myeloid cells, within tumor tissue compared to normal tissue, was demonstrated [48]. This suggests a great diversity in the immune landscape of individual tumors, which most likely is associated with the variety of individual tumor microenvironments, which can differ in inflammation, hypoxia and nutrient supply [49]. It could also help explain the differences in response to different immunotherapy regiments in breast cancers. Building further on the single cell approach, a single cell mass cytometry study on 144 breast tumor samples showed that PD-1+ T cells and PD-L1+ TAM were common to all breast cancer subtypes and linked increased PD-1 levels to higher T cell exhaustion rates. By looking at T cell subsets in ER+ and ER- breast cancer subtypes, they could determine that most of the analyzed ER- and some subsets of ER+ breast tumors had a higher infiltration of T-regs and high PD-1 + CTLA4+ expressing T cell subsets, indicating that these tumor subtypes could respond well to neoadjuvant immunotherapy utilizing checkpoint inhibitors [50]. This further exemplifies the power of single cell analysis approaches of patient samples, which can be used to determine what kind of therapy would have the best therapeutic effect considering the patient’s own immune landscape.

Taken together, TIL activity in the TME helps determine the outcome of some immunotherapies and partly explains why some individuals respond better to immunotherapy than others. Future immunotherapy approaches would gain a lot from looking at individual TIL setup in the TME and by using a personalized medicine approach tailoring the therapy to fit the patient. This would allow for manipulation of the dominant TIL subset through either activating effector cells to fight the tumor or inhibiting immunoinhibitory immune cells.

### 2.3. Tumor-Associated Macrophages (TAM)

Tumor-associated macrophages (TAM) are usually described as a macrophage subtype in the TME that displays an M2-like phenotype. They stimulate tumor progression by promoting tumor cell invasion, migration (MMPs), production of anti-inflammatory cytokines (TGF-beta, IL-10) and angiogenic factors (VEGF, TNF-alpha, HIF-1) [14,51]. The infiltration of TAM in the breast cancer stroma has been linked to more aggressive types of breast cancer (triple negative/basal like), cancer grade, tumor size and poor overall survival [20,52]. Therefore, TAM seem to play a defining role in the tumor microenvironment, and are thus a promising target for cancer treatment, in particular in combination approaches to overcome the solid tumor stroma.

Maturing macrophages respond effectively to cues in the environment in which they are recruited to by maturing into diverse subtypes with distinct functions [48,53,54]. Due to their plasticity, it has been suggested that the classical dichotomous characterization of macrophages is too simple and does not reflect the complexity of the tumor microenvironment [54,55]. Data supporting this comes from single cell analysis data of TAM characterized in breast cancer TME, where both M1 and M2 activated genes have been shown to be frequently expressed in the same cell in the breast cancer TME [48]. Moreover, it has been demonstrated that a subtype of pro-angiogenic TAM was coupled to poor clinical outcomes across multiple cancer types. This pro-angiogenic subtype could also be coupled to immunotherapy response in melanoma trials in which a better response to checkpoint inhibition could be seen when a low fraction of the pro-angiogenic subtype was present [54,56]. These findings overlap with findings from Wagner et al. in which PD-L1- expressing TAM were found across all analyzed breast cancer subtypes, suggesting that TAM have the ability to negatively influence the response to immunotherapy by checkpoint inhibition [50]. Furthermore, it has been suggested that TAM can interact with PD1 checkpoint inhibitors by Fcγ-receptor binding, leading to reduced interaction of checkpoint inhibition on CD8+ T cells, which in turn leads to a diminished therapeutic response of PD1 checkpoint inhibitors [57]. This further suggest immunoinhibitory and tumor protective properties of TAM and might explain the low response to checkpoint inhibition therapies in some patients. Moreover, when studying the TAM phenotype, Benner et al. found that in vitro-generated TAM displayed a high PD-L1 expression and an elevated gene expression of immunoinhibitory molecules, which was comparable to in vivo TAM [58]. These findings further underline the immunosuppressive function of TAM suggesting that combination therapies using PD-L1 checkpoint inhibitors might be a key strategy in the attempt to overcome the tumor stroma. However, a thorough design of the checkpoint antibodies needs to be considered to stop potential TAM interference by Fcγ-receptor binding.

TAM have further been implicated in the tumor protective effects that can be seen in standard breast cancer treatment regiments. Evidence points towards TAM influence against anti-mitotic agents by supporting tumor cells through survival signaling promoting anti-apoptotic responses. When depleting specific TAM subtypes, an increased Taxol- induced apoptosis was observed in mouse model intervention trials. These results implicate TAM ability to mediate chemoresistance against cytotoxic agents in the TME [59]. Previous research confirms this in which a chemotherapeutic resistance against Taxol, Doxorubicin and Etoposide could be conferred to TAM involvement [60].

Moreover, a high TAM infiltration in breast cancer tissue was positively correlated to tamoxifen resistance and higher EGFR expression in breast cancer tumors [61]. Later research established a reciprocal relationship between endocrine-resistant breast tumor cells and TAM linking endocrine resistance in tumor cells to CCL2 secretion promoted by TAM [62]. Furthermore, the tumor cells stimulated macrophages into an M2 phenotype, which in turn increased TAM CCL2 secretion promoting monocytes and macrophage accumulation in the TME creating a malignant positive feedback loop and endocrine resistance [61,62]. This reciprocal relationship underlines the effects of TAM in the breast stroma and its tumor-infiltrating properties, which can lead to therapy resistance.

Taken together, TAM appear to play a distinct tumor protective role in the tumor microenvironment, in which their ability to respond effectively to their environment seems to drive ECM modulation and tumor-protecting processes, thus driving malignancy and tumor progression. Therapies in which the local TAM population can be depleted or inhibited in the tumor microenvironment is thus a promising strategy in order to overcome the tumor stroma in future cancer treatment modalities.

## 3. Implications for Immunotherapy of Breast Cancer

Throughout this review, we have highlighted the most common components of the TME playing a defining role in the microenvironment of breast cancer that have implications for immunotherapy, and briefly discussed some strategies that can be used to overcome the microenvironment of a solid tumor. In this part of the review, we would like to summarize and discuss some immunotherapeutic approaches that are being worked on that might be of use for future breast cancer treatment modalities that can overcome the TME in breast cancer.

### 3.1. Antibodies

The only immunotherapy currently in use in breast cancer treatment are based on antibodies that target proliferation pathways such as HER2 in breast cancers or immunomodulators such as checkpoint inhibitors (CPI) that target TIL in the breast cancer TME. [3]. Recently, the antibody Sacituzumab targeting the TROP-2 pathway in triple-negative breast cancer was approved expanding the targeted antibody treatment options beyond HER2 dependent breast cancers [63,64]. Checkpoint inhibitors act on either the PD1/PD-L1 axis or CTLA-4 on T cells which reduces the interaction between cancer cells and immune effector cells, preventing the tumor cells from immune escape [65]. They are commonly administered with other forms of therapies such as radiotherapy or chemotherapy. CPIs acting on PD1 are pembrolizumab and nivolumab, whereas PD-L1 inhibitors are atezolizumab, durvalamab and avelumab [66]. CTLA-4 inhibitors used are ipilimumab and tremelimumab [66]. The effectiveness and viability of CPIs in breast cancer treatment have been extensively reviewed elsewhere and will not be focused on here [65,66,67,68]. Nevertheless, CPIs are a powerful tool that can be used in future combination therapies that could have synergistic effects when combined with cellular immunotherapies. This could help to overcome the tumor microenvironment by making the cellular therapy response more durable and persistent.

There are immunotherapeutic strategies being worked on that target the TME specifically, but none of these have been used to target the tumor microenvironment of breast cancer in a clinical setting. However, there are immunotherapies targeting key components of the TME that are being worked on. Cancer therapy approaches targeting CAF for example have been promising in pre-clinical testing, but these have been hard to translate into the clinic [69]. Current immunotherapeutic approaches targeting CAF directly or indirectly are blocking antibodies targeting CAF actions or pathways, such as Simtuzumab targeting the enzyme LOXL2 [70] and Pamrevlumab targeting the connective tissue growth factor CTGF [71,72]. Additionally, combination therapies involving small inhibitor molecules and blocking antibodies such as Galunisertib with Durvalumab targeting TGF-beta signaling in the TME [73] or RO687428 + Atezolizumab directly targeting the CAF marker FAP are also being worked on [74].

The tumor protective effects of TAM in the TME could be counteracted by strategies involving blocking antibodies in order to deplete or inactivate TAM in the TME. Immunotherapy approaches targeting TAM are for example targeting CSF-1R with the blocking antibody Emactuzumab that leads to TAM depletion in the TME by affecting proliferative signaling in the TAM population [75]. Recently, Binnewies et al. described that the surface molecule TREM2 was being enriched in the TAM population of PD1-resistant tumors. Through targeting of TREM2 with an anti-TREM2 monoclonal antibody, they could show a depletion of TAM in the TME and an increase in anti-tumor activity [76]. This shows that new antibody designs targeting TAM could lead to less TAM activity in the TME and more effective immunotherapies in the treatment of solid tumors.

### 3.2. Nanoparticles

An emerging approach targeting TME components are nanoparticles which can be used as a combination approach together with immunotherapies. Nanoparticles as a drug delivery system can be more easily absorbed and penetrate the TME better than traditional forms of drug delivery and show prolonged retention times [77]. Due to their smaller size and absorbability, the tumor protective effects of TAM and CAF could be surpassed. The viability of nanoparticles in a breast cancer setting has been demonstrated by Ramesh et al., in which they managed to repolarize M2 macrophages into M1 macrophages by using supramolecular nanoparticles containing inhibitors targeting CSF1R and MAPK pathways of macrophages [78]. This strategy would allow for targeted depletion of TME components such as TAM and CAF, which diminishes the tumor protective effects of the TME, allowing for immunotherapies to be more effective. Recently, Yang et al. have extensively reviewed the application of nanoparticles in cancer immunotherapy and have highlighted the main strategies being worked on in the nanoparticle field that would disrupt main TME functions [77].

### 3.3. Cellular Immunotherapy

Cellular immunotherapy has gained traction during the last decade with a large potential to expand solid tumor treatment options. The interest in the field is largely due to the progress that has been made in synthetic biology and the success of autologous CAR T-cell therapy against hematological malignancies [79]. However, this is a developing field, and there are still a few challenges and bottlenecks to overcome. One of these challenges is the translation into a solid tumor setting [7,80]. In order to improve next generation cell therapies, strategies being worked on include increasing their viability by exploring different cell sources such as iPSC, finding better molecular targets, improving gene modifications and increasing efficiency through stimulating molecules and combination therapy approaches [80,81,82,83]. An example of a strategy that could overcome the TME is to combine two different targeted cell therapies, in which one specifically targets TME components depleting the TME and the other targeting the tumor cells to increase anti-tumor efficiency. The viability of such a strategy against solid tumors was recently demonstrated by Rodriguez-Garcia et al. in which they managed to specifically deplete TAM by the sequential administration of TAM targeting T-cells followed by cancer targeting T-cells, which lead to tumor regression and extended survival in mouse models [84]. Even though cellular immunotherapy shows good functionality in pre-clinical breast cancer models [85,86,87], they have to date not been proven a monotherapy solution to treating breast cancer. In current breast cancer clinical trials, they are and most probably will be reliable upon combination approaches in order to overcome the TME and have a lasting therapeutic response such as the ones that have been seen in hematological malignancies. To demonstrate this, we searched for recent active trials involving cellular immunotherapy utilizing immune effector cells targeting breast cancer focusing on recently added studies in early phases (Table 1). We identified 17 ongoing trials, and 10 out of 17 include some form of combination approach using either chemotherapy or checkpoint inhibitors. Two of these trials are testing CPIs with cellular therapy, whereas four are testing CPIs and chemotherapy combined with cellular therapy and four are testing chemotherapy and cellular therapy. Nevertheless, all the combination approaches broadly weaken the TME either through the influence of cell dividing processes through chemotherapy or immunomodulation using CPIs. This is done either before or at the same time as the cellular therapy is administered [88,89,90,91,92,93,94,95,96]. However, chemotherapy and CPIs are administered systematically, which widens the side effect profile. Combining cellular therapies with therapies targeting the TME components specifically, such as the antibodies mentioned above, might lead to a better response. Thus far, there is a lack of specific targeting of TME components but considering the growing understanding of the TME influence in cancer treatment and the ongoing research in the field, this is bound to change in the future. The best cellular immunotherapeutic strategy and form of administration for breast cancers that will lead to a durable response is yet to be determined.

## 4. Concluding Remarks

Overcoming the microenvironment in solid tumor immunotherapy is not a trivial task due to the multifactorial processes and components of the TME that both protects the tumor and drives its progression. As yet, there are no “magic bullet” solutions to this problem and a holistic approach against major components of the TME needs to be taken into account.

A promising addition to cancer research is the more widespread use of single-cell sequencing, transcriptomic analyses and NGS approaches characterizing the different responses to immunotherapy. This will allow for a more personalized therapy approach allowing the optimal treatment modalities to be combined to fit each individual patient and thus improve the response of immunotherapies.

In future immunotherapy modalities, we believe that combination approaches utilizing cell therapy, CPI and nanoparticles might become one of the more viable immunotherapy strategies targeting breast cancer and the tumor microenvironment. This combination approach would deplete or reprogram the components of the TME by using nanoparticles and target cancer cells with the homing effects of immunotherapies. Using the strengths of both therapy strategies combined would potentially increase effectivity and the persistence of immunotherapies.

## Figures and Tables

**Figure 1 cancers-14-03178-f001:**
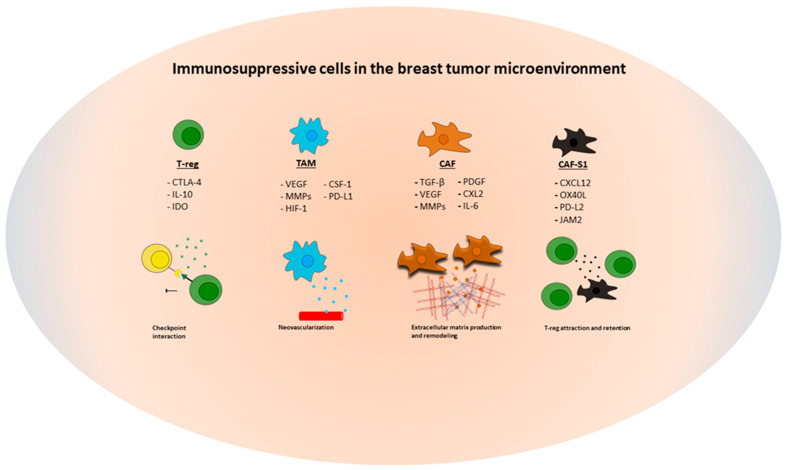
Illustration of the immunosuppressive cell types in the breast tumor microenvironment discussed in this review with a selection of the most common molecules and mechanisms of action that promotes tumor growth. Abbreviations used for cell types and signaling molecules: CAF (cancer-associated fibroblasts); CAF-S1 (cancer-associated fibroblasts—subtype 1); TAM (tumor-associated macrophages); T-reg (regulatory T-cells); CSF-1 (colony-stimulating factor 1); CTLA4 (cytotoxic T-lymphocyte-associated); CXCL (C-X-C motif chemokine); HIF-1 (hypoxia-inducible factor 1); IDO (Indoleamine 2,3-dioxygenase); IL (interleukin); JAM2 (junctional adhesion molecule 2); MMP (matrix-metalloproteinase); OX40L (tumor necrosis factor ligand); (PD-L1/2 (programmed death ligand 1/2); PDGF (platelet-derived growth factor); TGF (transforming growth factor); VEGF (vascular endothelial growth factor).

**Table 1 cancers-14-03178-t001:** Summary of recent active clinical trials using cellular immunotherapy against breast cancer as a monotherapy or in combination approaches with checkpoint inhibitors or chemotherapy. Clinicaltrials.gov, accessed March 2022, search terms used were “cellular immunotherapy” and” breast cancer”, with the search criteria focusing on recently added studies in early phases using cellular immunotherapy against breast cancer excluding vaccination approaches.

Study Title	NCT	Interventions	Cell Target	Phase
HER2-CAR T Cells in Treating Patients With Recurrent Brain or Leptomeningeal Metastases	NCT03696030 [97]	Biological: Chimeric Antigen Receptor T-Cell Therapy	HER2	Phase 1; Recruiting
Autologous huMNC2-CAR44 T Cells for Breast Cancer Targeting Cleaved Form of MUC1 (MUC1*)	NCT04020575 [98]	Biological: huMNC2-CAR44 CAR T cells Biological: huMNC2-CAR44 CAR T cells @ RP2D	MUC1	Phase 1; Recruiting
EpCAM CAR-T for Treatment of Nasopharyngeal Carcinoma and Breast Cancer	NCT02915445 [99]	Biological: CAR-T cells recognizing EpCAM	EpCAM	Phase 1; Recruiting
Genetically Engineered Cells (MAGE-A1-specific T Cell Receptor-transduced Autologous T-cells) and Atezolizumab for the Treatment of Metastatic Triple Negative Breast Cancer, Urothelial Cancer, or Non-small Cell Lung Cancer	NCT04639245 [88]	Biological: MAGE-A1-specific T Cell Receptor-transduced Autologous T-cells Biological: PD1 Inhibitor Drug: Atezolizumab Drug: Fludarabine Drug: Cyclophosphamide	MAGE-A1	Phase 1/2; Recruiting
T-Cell Therapy for Advanced Breast Cancer	NCT02792114 [100]	Biological: Mesothelin-targeted T cells Drug: Cyclophosphamide Drug: AP1903	Mesothelin	Phase 1; Active, not recruiting
BATs in Patients With Breast Cancer and Leptomeningeal Metastases	NCT03661424 [101]	Drug: HER2 BATs	n.a	Phase 1; Recruiting
RAPA-201 Therapy of Solid Tumors	NCT05144698 [90]	Biological: RAPA-201 Rapamycin Resistant T Cells Drug: Chemotherapy Prior to RAPA-201 Therapy	n.a	Phase 2; Recruiting
CAR-T Intraperitoneal Infusions for CEA-Expressing Adenocarcinoma Peritoneal Metastases or Malignant Ascites (IPC)	NCT03682744 [102]	Biological: anti-CEA CAR-T cells	CEA	Phase 1; Active, not recruiting
Her2-BATS and Pembrolizumab in Metastatic Breast Cancer	NCT03272334 [91]	Drug: HER2 BATs with Pembrolizumab	HER2	Phase 1/2; Recruiting
Malignant Pleural Disease Treated With Autologous T Cells Genetically Engineered to Target the Cancer-Cell Surface Antigen Mesothelin	NCT02414269 [89]	Genetic: iCasp9M28z T cell infusions Drug: Cyclophosphamide Drug: Pembrolizumab	Mesothelin	Phase 1/2; Active, not recruiting
A Study to Investigate LYL797 in Adults With Solid Tumors	NCT05274451 [103]	Biological: LYL797	ROR1	Phase 1; Not yet recruiting
C7R-GD2.CART Cells for Patients With Relapsed or Refractory Neuroblastoma and Other GD2 Positive Cancers (GAIL-N)	NCT03635632 [92]	Genetic: C7R-GD2.CART cells Drug: Cyclophosphamide Drug: Fludarabine	GD2	Phase 1; Recruiting
A Study of Gene Edited Autologous Neoantigen Targeted TCR T Cells With or Without Anti-PD-1 in Patients With Solid Tumors	NCT03970382 [93]	Biological: NeoTCR-P1 adoptive cell therapy Biological: Nivolumab Biological: IL-2	neoepitope (neoE)	Phase 1; Active, not recruiting
FT500 as Monotherapy and in Combination With Immune Checkpoint Inhibitors in Subjects With Advanced Solid Tumors	NCT03841110 [94]	Drug: FT500 Drug: Nivolumab Drug: Pembrolizumab Drug: Atezolizumab Drug: Cyclophosphamide Drug: Fludarabine Drug: IL-2	n.a	Phase 1; Recruiting
Immunotherapy Combined With Capecitabine Versus Capecitabine Monotherapy in Advanced Breast Cancer	NCT02491697 [104]	Biological: DC-CIK Immunotherapy Drug: Capecitabine Monotherapy and Combination	n.a	Phase 2; Active, not recruiting
A Study of DC-CIK Immunotherapy in the Treatment of Solid Tumors	NCT04476641 [95]	Other: CELL	n.a	Phase 2; Recruiting
Immunotherapy Using Tumor-infiltrating Lymphocytes for Patients With Metastatic Cancer	NCT01174121 [96]	Biological: Young TIL Drug: Aldesleukin Drug: Cyclophosphamide Drug: Fludarabine Drug: Pembrolizumab	n.a	Phase 2; Recruiting

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
