# Peer review of "Implications for Immunotherapy of Breast Cancer by Understanding the Microenvironment of a Solid Tumor"

_cancers, 2022, doi:10.3390/cancers14133178_

Round 1

Reviewer 1 Report

This review is a clear summary of the cell types present in breast tumor microenvironment and the immunotherapeutic related strategies developed until now. 

I consider it can be published in the present version.

Author Response

Review 1 comments and replies

This review is a clear summary of the cell types present in breast tumor microenvironment and the immunotherapeutic related strategies developed until now. 

I consider it can be published in the present version.

  • We thank the reviewer for the comment

Reviewer 2 Report

Thank you for letting me read this interesting paper. I have no comments really besides a minor detail, page 2 line 73 (Sahi, ref), is this referring to Sahai9 ?

Author Response

Review 2 comments and replies

Thank you for letting me read this interesting paper. I have no comments really besides a minor detail, page 2 line 73 (Sahi, ref), is this referring to Sahai9 ?

  • We thank the reviewer for the comment and we have fixed the reference.

Reviewer 3 Report

This review manuscript by Franzen et al. addresses the timely topics of the tumor microenvironment (TME) and cancer immunotherapy with focus on breast cancer.

The abstract promises to “summarize current understanding of the microenvironment ….. giving implications for immunotherapeutic strategies”. The manuscript touches on many of these vast and complex aspects but fails to convey a comprehensive picture of the field. The review should be either significantly expanded or limited to a more narrow focus.

Major Points.

1) It is unclear who is the target audience. The authors do not provide background information and definitions on immunotherapy or breast cancer subtypes. However, readers with knowledge in both of these areas will not gain substantially new insights. Thus, one or two sections should be added to provide such background information. The authors may want to consider organizing the manuscript by type of immunotherapy.

2) TIL’s are generally defined as “lymphocytes and plasma cells” (PMID: 34853355). The authors have focused only on T-cells and touched on NK cells without their functional implications. The term TILs should be avoided when only referring to T-cell subtypes. Consequently, much of section on TILs is confusing as plasma cells/B cells are completely ignored.

3) The authors declare that the article focuses on the “most abundant components” of the TME (Line 52). Instead, it appears that the focus is on components with the “most abundant representation in the literature”. For example, section 2.3. is almost exclusively on M2 TAMs, while recent studies have documented a wide variety of myeloid cell types including neutrophil subtypes in the TME.

4) The manuscript preparation was sloppy: For example, Line 73 states “Sahi, ref”, suggesting that a reference should be inserted. Lines 163-165 appear to contain someone’s comments on the manuscript: “This section may be divided by subheadings. It should provide a concise and precise description of the experimental results, their interpretation, as well as the experimental conclusions that can be drawn.” 

5) Table 1 is not very helpful as most approaches are based on CAR-T cells and targets on tumor cells and therefore do not impinge on the TME. The authors list 17 “active” trials at early stage. What were the search terms? “Breast” and “immunotherapy” as search terms on the same site (clinicaltrials.gov) yields 352 studies, 192 “active”, 90 completed, and 36 have posted results. The authors should more clearly define and also broaden their selection criteria. Information on and discussion of completed trials with results could provide more insights.

6) Section 3 lacks structure and overall, there should be more on Synthesis and Perspectives, not just global statements on how targeting the TME will improve therapies in the future.

Minor Points:

Title: Consider deleting “of a solid cancer” from the title.

Figure 1: Focus on either markers or functions, or consistently state both. A table might be more suitable than an image. The complexity of CAF subtypes should be included as well.

Line 8: Breast cancer is not a “growing problem” because life expectancy is increasing steadily with more and better treatment options. 

Line 44: replace “this form of cancer” with more specific information (what form of cancer?).

Line 86: provide citation for TGFbeta activity on CTLs and NK cells.

Line 94: Proximity of CAFs and tumor cells does not by itself support the conclusion that CAFs are initiators of tumor development.  

Line 27-28: it is incorrect to say that breast cancer “leads” the global mortality statistics, since it is “only” the fifth-leading cause of cancer mortality.

Line 181-187: bi-direction and tri-directional antibodies are mentioned without explanations on what they are. This is just one example for lack of background.

Line 310 Smituzumab etc.: state the specific targets when drugs are named.

Line 342: The “smaller size” of nanoparticles is mentioned here and subsequently. “Smaller” compared to what? Nanoparticles are larger than the drugs they deliver!

Line 345: why state “active inhibitors” and not just “inhibitors”? What are inactive inhibitors?

Line 354-355: Therapies do not address incidence rates since they are treatments and not prevention measures.

Line 355 “reignited”: explain or delete.

Line 381: add citation of original research article.

Line 388-390: Research approaches are erroneously listed as “cancer-fighting arsenal”

Lastly, the manuscript would benefit from professional editing. Some examples:

  • Line 27: convert “has long been” to “is”.
  • Lines 148-149, 194, 240, 245: examples of verb and subject not matching.
  • Line 159: too many verbs.
  • Line 305: “cancer modalities” should be “cancer treatment modalities”.
  • Line 347-352: “diminishing the protective effects…..effective immunotherapy” is repeated twice.
  • Rather than starting sentences or even paragraphs with just “This” (or “it”), add a noun to specify what “this” refers to.
  • The use of the term CAF versus CAFs is inconsistent.

Author Response

Review 3 comments and replies

This review manuscript by Franzen et al. addresses the timely topics of the tumor microenvironment (TME) and cancer immunotherapy with focus on breast cancer.

The abstract promises to “summarize current understanding of the microenvironment ….. giving implications for immunotherapeutic strategies”. The manuscript touches on many of these vast and complex aspects but fails to convey a comprehensive picture of the field. The review should be either significantly expanded or limited to a more narrow focus.

  • In order to clarify our review we now concentrate on cellular therapy

Major Points.

1) It is unclear who is the target audience. The authors do not provide background information and definitions on immunotherapy or breast cancer subtypes. However, readers with knowledge in both of these areas will not gain substantially new insights. Thus, one or two sections should be added to provide such background information. The authors may want to consider organizing the manuscript by type of immunotherapy.

  • We now include background on breast cancer subtypes and immunotherapy (32-59)

2) TIL’s are generally defined as “lymphocytes and plasma cells” (PMID: 34853355). The authors have focused only on T-cells and touched on NK cells without their functional implications. The term TILs should be avoided when only referring to T-cell subtypes. Consequently, much of section on TILs is confusing as plasma cells/B cells are completely ignored.

  • We have expanded the TIL section and include discussion on B-cells and plasma cells (210-214).

3) The authors declare that the article focuses on the “most abundant components” of the TME (Line 52). Instead, it appears that the focus is on components with the “most abundant representation in the literature”. For example, section 2.3. is almost exclusively on M2 TAMs, while recent studies have documented a wide variety of myeloid cell types including neutrophil subtypes in the TME.

  • We have amended the manuscript to clarify this point (69-71)

4) The manuscript preparation was sloppy: For example, Line 73 states “Sahi, ref”, suggesting that a reference should be inserted. Lines 163-165 appear to contain someone’s comments on the manuscript: “This section may be divided by subheadings. It should provide a concise and precise description of the experimental results, their interpretation, as well as the experimental conclusions that can be drawn.” 

  • We have now corrected and proofread the manuscript.

5) Table 1 is not very helpful as most approaches are based on CAR-T cells and targets on tumor cells and therefore do not impinge on the TME. The authors list 17 “active” trials at early stage. What were the search terms? “Breast” and “immunotherapy” as search terms on the same site (clinicaltrials.gov) yields 352 studies, 192 “active”, 90 completed, and 36 have posted results. The authors should more clearly define and also broaden their selection criteria. Information on and discussion of completed trials with results could provide more insights.

  • We have added a discussion to the reason why the table was included, the search words were focusing on recent cellular therapies added to get the feeling of what are currently being worked on in the field. Point being that none of the ongoing trials are currently targeting the TME specifically, however, a majority of them recognize the TME as a problem and combines the cellular therapy with some form of TME weakening step.

6) Section 3 lacks structure and overall, there should be more on Synthesis and Perspectives, not just global statements on how targeting the TME will improve therapies in the future.

  • We have now expanded Section 3

Minor Points:

Title: Consider deleting “of a solid cancer” from the title.

  • We have considered it but we decided to keep the current title since this review also covers solid tumor microenvironment.

Figure 1: Focus on either markers or functions, or consistently state both. A table might be more suitable than an image. The complexity of CAF subtypes should be included as well.

  • We have amended figure 1 to the reviewers comment and focused on the immunosuppressive subset of cells that can be found in the TME.

Line 8: Breast cancer is not a “growing problem” because life expectancy is increasing steadily with more and better treatment options. 

  • fixed

Line 44: replace “this form of cancer” with more specific information (what form of cancer?).

  • fixed

Line 86: provide citation for TGFbeta activity on CTLs and NK cells.

  • fixed

Line 94: Proximity of CAFs and tumor cells does not by itself support the conclusion that CAFs are initiators of tumor development.  

  • fixed

Line 27-28: it is incorrect to say that breast cancer “leads” the global mortality statistics, since it is “only” the fifth-leading cause of cancer mortality.

  • fixed

Line 181-187: bi-direction and tri-directional antibodies are mentioned without explanations on what they are. This is just one example for lack of background.

  • fixed

Line 310 Smituzumab etc.: state the specific targets when drugs are named.

  • fixed

Line 342: The “smaller size” of nanoparticles is mentioned here and subsequently. “Smaller” compared to what? Nanoparticles are larger than the drugs they deliver!

  • Fixed

Line 345: why state “active inhibitors” and not just “inhibitors”? What are inactive inhibitors?

  • Fixed

Line 354-355: Therapies do not address incidence rates since they are treatments and not prevention measures.

  • True, its removed

Line 355 “reignited”: explain or delete.

  • deleted

Line 381: add citation of original research article.

  • done

Line 388-390: Research approaches are erroneously listed as “cancer-fighting arsenal”

  • fixed

Lastly, the manuscript would benefit from professional editing. Some examples:

  • Line 27: convert “has long been” to “is”.
  • Lines 148-149, 194, 240, 245: examples of verb and subject not matching.
  • Line 159: too many verbs.
  • Line 305: “cancer modalities” should be “cancer treatment modalities”.
  • Line 347-352: “diminishing the protective effects…..effective immunotherapy” is repeated twice.
  • Rather than starting sentences or even paragraphs with just “This” (or “it”), add a noun to specify what “this” refers to.
  • The use of the term CAF versus CAFs is inconsistent.

It has been corrected

Reviewer 4 Report

This is an interesting review, however, it need some extensive editing and language review. There are multiple mistakes and a lack of proper comma rules. The sections on the different cell types are too long and detailed. Rather than giving an overall synopsis, a lot of individual studies are cited. For example, in the TAM section, lines 263 to 268 could be omitted. On the other hand, I think more focus on the individuality of TME between patients and the impact of this individuality on immune therapy response would be interesting.

  • some of the references are missing, some are the names of the authors (for example line 73), and some are numbers.
  • line 39: I think it is incorrect to suggest that the TME transforms healthy tissue into malignant tumor. Most tumors develop due to genomic aberrations which have nothing or little to do with the TME. This makes it sound as if the TME is causing the cancer, which is incorrect.
  • the last part of secsion 2.1 seems to be an instruction on how to compose the manuscript, and not a part of the article itself. Please remove.
  • line 92: I think the evidence to bolster this statement is circumstantial at best, I would not state that CAF are involved in tumor progression.
  • line 82: what ist the difference between ER- and triple-negative tumors? Should this read ER+?
  • the section of the implications of the TME for immunotherapy should be expanded, because this is really the most interesting part.
  • table 1 could be omitted. Why show 17 ongoing trials on cellular immunotherapy, but no studies on any of the other therapeutic concepts mentioned?
  • line 354: I think proposing that cellular immunotherapy is a solution to rising cancer rates is a bit far fetched. Maybe it is a solution in for cancer therapy, but I don't see how cellular immunotherapy could prevent cancer and lower the incidence.

Author Response

Reviewer 4 comments and replies:

General comments:

This is an interesting review, however, it need some extensive editing and language review. There are multiple mistakes and a lack of proper comma rules. The sections on the different cell types are too long and detailed. Rather than giving an overall synopsis, a lot of individual studies are cited. For example, in the TAM section, lines 263 to 268 could be omitted. On the other hand, I think more focus on the individuality of TME between patients and the impact of this individuality on immune therapy response would be interesting.

  • We agree with the reviewer and have amended the manuscript according to the comment shortening parts of the review and expanded the final section (lines 382-389, 401-413,457-470 478-488).

  • some of the references are missing, some are the names of the authors (for example line 73), and some are numbers.

We have proofread the review and fixed the references.

  • line 39: I think it is incorrect to suggest that the TME transforms healthy tissue into malignant tumor. Most tumors develop due to genomic aberrations which have nothing or little to do with the TME. This makes it sound as if the TME is causing the cancer, which is incorrect.

Very good point, we did not mean it to come out like that and have changed it (lines 66-68).

  • the last part of section 2.1 seems to be an instruction on how to compose the manuscript, and not a part of the article itself. Please remove.

The section has been removed, it was overseen while editing the manuscript to fit the given manuscript template.

  • line 92: I think the evidence to bolster this statement is circumstantial at best, I would not state that CAF are involved in tumor progression.

We agree with the reviewer and have clarified this point (lines 132-133)

  • line 182: what is the difference between ER- and triple-negative tumors? Should this read ER+?

Typing error, it has been fixed. We now include a section on breast cancer types for clarity (lines 37-43).

  • the section of the implications of the TME for immunotherapy should be expanded, because this is really the most interesting part.

We agree with the reviewer and have expanded this section.

  • table 1 could be omitted. Why show 17 ongoing trials on cellular immunotherapy, but no studies on any of the other therapeutic concepts mentioned?

 With the expansion of the last section the table might make more sense now?

  • line 354: I think proposing that cellular immunotherapy is a solution to rising cancer rates is a bit far fetched. Maybe it is a solution in for cancer therapy, but I don't see how cellular immunotherapy could prevent cancer and lower the incidence.

True, it has been fixed.

Round 2

Reviewer 3 Report

The authors have addressed the main concerns in part by providing more detailed background and by being more explicit about definitions and what the review is trying to accomplish.

Some minor issues remain:

1) Figure 1 does not include neutrophil variants of MDSCs for example. The legend should be changed from “Illustration of the most common immunosuppressive cell types” to something to the effect of “Illustration of the immunosuppressive cell types that are being discussed in this review”.

2) The introductory remarks on immunotherapy are still confusing. Antibodies against cell surface receptors can be used to inhibit the activity of the receptors, as opposed to engaging the immune system (anti-HER2 for example). This distinction should be clarified and a reference review should be cited that outlines the definitions and various modalities of immunotherapies.

3) Search terms/criteria for Table 1 are still missing.

Author Response

The authors have addressed the main concerns in part by providing more detailed background and by being more explicit about definitions and what the review is trying to accomplish.

Some minor issues remain:

1) Figure 1 does not include neutrophil variants of MDSCs for example. The legend should be changed from “Illustration of the most common immunosuppressive cell types” to something to the effect of “Illustration of the immunosuppressive cell types that are being discussed in this review”.

  • Has been changed to “illustration of the immunosuppressive cell types discussed in this review” (Line 94-96).

2) The introductory remarks on immunotherapy are still confusing. Antibodies against cell surface receptors can be used to inhibit the activity of the receptors, as opposed to engaging the immune system (anti-HER2 for example). This distinction should be clarified and a reference review should be cited that outlines the definitions and various modalities of immunotherapies.

  • We have clarified this and we have added the corresponding citations (line48-56):

“In breast cancer, current immunotherapies are focused around antibodies targeting molecular receptors on the breast cancer cell surface or targeting the tumor infiltrating immune cell subset in the tumor microenvironment. The former treatment leads to receptor blockade inhibiting proliferative pathways, such as HER2 [5]. The latter treatment makes the immune cells more potent in finding and eliminating tumor cells or preventing the immune cells from being inactivated by the tumor cells [3,4,6]”

3) Search terms/criteria for Table 1 are still missing.

  • Search criteria and terms have been added in the text (line 483-485) and has been further specified in the table description (line 505-508).

Reviewer 4 Report

In my opinion, the introduction for the different TME cells types is still too extensive. I would focus more on available therapies and less on introducing the cell types. I also don't see the benefit of table 1 (as noted before). Why put so much focus on cellular immunotherapy, which currently plays no role at all in the treatment of breast cancer. Additionally, the language also still needs revising.

Author Response

In my opinion, the introduction for the different TME cells types is still too extensive. I would focus more on available therapies and less on introducing the cell types. I also do not see the benefit of table 1 (as noted before). Why put so much focus on cellular immunotherapy, which currently plays no role at all in the treatment of breast cancer. Additionally, the language also still needs revising.

  • To our knowledge there are not many available immunotherapies other than the ones already mentioned in the review that specifically target the TME. Our intent is to give a broader understanding of the tumor microenvironment by focusing on the best understood and represented in cell types in the literature. We try to do this by deep diving into these cell types and then to discuss possible ways on overcoming the problem.

  • We have added a discussion to the reason why the table was included and further specified the search term and criteria’s. The search criteria’s were focusing on recent cellular therapies in early phases to exemplify what are currently being developed and focused on in the field. The point being that none of the ongoing trials are currently targeting the TME specifically, however, a majority of them recognize the TME as a problem and combines the cellular therapy with some form of TME weakening step.

  • You are correct, right now, there are less cellular therapies being tested in breast cancer, but there are several studies using TILs in breast cancer and in our opinion in the future improved cellular therapies will be included in protocols in the treatment of breast cancer. Especially, cellular therapies using improved next generation CAR-vectors, offer the possibility to fight the immunosuppressive TME.

  • The manuscript have been further corrected and proofread.